# Repeat controlled human *Plasmodium falciparum* infections delay bloodstream patency and reduce symptoms

Patricia Ferrer [1,2], Andrea A. Berry [3], Allison N. Bucsan [1,2], Surendra K. Prajapati [1,2], Karthik Krishnan[1,2], Michelle C. Barbeau[1,2], David M. Rickert[1,2], Sandra Mendoza Guerrero[1,2], Miho Usui [1,2], Yonas Abebe[4], Asha Patil[4], Sumana Chakravarty[4], Peter F. Billingsley [4], Faith Pa'ahana-Brown[3], Kathy Strauss[3], Biraj Shrestha[3], Effie Nomicos[5], Gregory A. Deye[5], B. Kim Lee Sim[4], Stephen L. Hoffman [4], Kim C. Williamson [1,6] & Kirsten E. Lyke [3,6]

Resistance to clinical malaria takes years to develop even in hyperendemic regions and sterilizing immunity has rarely been observed. To evaluate the maturation of the host response against controlled repeat exposures to *P. falciparum* (Pf) NF54 strain-infected mosquitoes, we systematically monitored malaria-naïve participants through an initial exposure to uninfected mosquitoes and 4 subsequent homologous exposures to Pf-infected mosquitoes over 21 months (*n* = 8 males) (ClinicalTrials.gov# NCT03014258). The primary outcome was to determine whether protective immunity against parasite infection develops following repeat CHMI and the secondary outcomes were to track the clinical signs and symptoms of malaria and anti-Pf antibody development following repeat CHMI. After two exposures, time to blood stage patency increases significantly and the number of reported symptoms decreases indicating the development of clinical tolerance. The time to patency correlates positively with both anti-Pf circumsporozoite protein (CSP) IgG and CD8 + CD69+ effector memory T cell levels consistent with partial pre-erythrocytic immunity. IFNγ levels decrease significantly during the participants' second exposure to high blood stage parasitemia and could contribute to the decrease in symptoms. In contrast, CD4-CD8 + T cells expressing CXCR5 and the inhibitory receptor, PD-1, increase significantly after subsequent Pf exposures, possibly dampening the memory response and interfering with the generation of robust sterilizing immunity.

Malaria continues to be among the world's most impactful infectious diseases accounting for the death of 627,000 people in 2021[1]. Unlike many viral infections, children usually have repeated clinical episodes characterized by fever, chills, headache, myalgia, and nausea. By adulthood, the majority of individuals living in endemic areas are protected against severe illness, but may continue to carry the parasite without apparent symptoms[2]. The reasons for this gradual tolerance to the parasite are not well established. Vaccine responses have also been found to be lower in endemic compared with naïve populations, complicating the development of effective vaccines[3] and highlighting

**Fig. 1 | Repeat CHMI time course. A** Over the course of two years, healthy participants were first challenged with 5 uninfected mosquitoes (mock = m ($n = 6$)) followed by 3 ($n = 8$) or 4 ($n = 5$) challenges with 5 Pf-NF54-infected mosquitoes (CHMI) (salivary gland count >2 (100–1000 sporozoites)). **B** Blood sample collection schedule after each of the challenges. Prior to the challenge (0), on days 1, 6, 8, treatment day (DRx) and DRx+7 samples were collected for plasma and PBMC isolation and cryopreservation (blue arrow). Every day from D6 until the clearance of parasites after treatment a sample was taken for quantitative PCR and blood smear to monitor parasitemia (blue dot). On DRx+7 plasmablasts were isolated from PBMCs collected by apheresis (*).

**Fig. 2 | Repeat CHMI Clinical Trial CONSORT Diagram.** Twenty-seven participants were enrolled over the course of the study to either receive sequential controlled human malaria infections (CHMI) (14) or a single CHMI (13) as controls for mosquito infectivity. Of the 14 participants enrolled to received repetitive CHMIs 10 completed the mock CHMI, and an additional 4 were enrolled at CHMI 2 as replacements for the 4 original participants that withdrew. In total, 5 participants received 4 infectious CHMIs and 3 participants received 3 infectious CHMI and were included in the final analysis.

the need to understand the molecular basis of the immune response to facilitate the design of more effective immunization strategies.

In response to natural malaria exposure, Pf-specific antibodies and memory B cells have been shown to increase over time[2] and passive transfer experiments clearly demonstrate that IgG isolated from adults can reduce parasitemia in children[4,5]. In regions with intense, seasonal malaria transmission, anti-parasite antibodies are generated after each malaria season, but the responses to many antigens are short-lived and decrease significantly in the absence of transmission during the dry season[2]. In subsequent years there is a small increase in the overall anti-parasite Ig titer that plateaus after age 18 years when most individuals are protected against severe clinical episodes[2]. Similarly, in the absence of treatment, syphilis patients undergoing *P. falciparum* malaria fever therapy in the 1940–60 s continued to have waves of recrudescent parasites about every 20 days for 4 months and remained susceptible to reinfection with the same parasite strain[6–8]. Peak parasitemia decreased over time until it was no longer detectable by blood smear, but it was unclear if participants were ever sterilely protected given the lack of sensitive nucleic acid detection methods at the time.

The factors that contribute to the gradual immune response remain an active area of investigation. Individuals living in malaria-endemic areas have been reported to have fewer circulating immature B cells and naïve B cells as well as an expanded population of activated and atypical memory B cells relative to malaria naïve individuals[9], while the levels and responsiveness of γδ2 T cells decline with each subsequent infection[10]. In addition to increased Ig levels over time, adults living in endemic areas also have a higher parasitemia threshold to trigger symptoms[11]. More recent work has implicated a decrease in monocyte activation and proinflammatory cytokine secretion in this reduction of symptoms[12,13].

The clinical trial reported here monitors the maturation of the immune response through three to four sequential exposures to Pf-infected mosquitoes. A detailed analysis of the gamma delta T cell response of the participants has already been reported[14,15]. Here, we describe the clinical and humoral immune response as well as the B, CD4 +, and CD8 + T cell responses. Prior to this longitudinal clinical trial, the well-established controlled human malaria infection (CHMI) model utilizing Pf-infected mosquitoes to initiate an infection at a defined time with a known parasite isolate has primarily been used to test vaccine efficacy[16] and allowed analysis of the immune response to a single, initial Pf exposure[17]. Intraerythrocytic parasites can usually be detected by Giemsa-stained smear within 9–12 days in naïve

participants and are often accompanied by fever and symptoms, which are also the hallmarks of symptomatic malaria infections in the field[16]. This prior in vivo work coupled with in vitro data have suggested that there is an initial proliferation of γδ T-cells, which has been associated with IFNγ secretion[18], and this was thought to stimulate monocytes to secrete TNF, IL-6, and IL-1, which can trigger fever[17,19]. Monocytes also secrete chemokines and IL-12, which can stimulate natural killer (NK) cells to release IFNγ[20] further enhancing the inflammatory response that in a rodent malaria model has been shown to dampen the development of B cells in the germinal centers[21]. Some participants also have spikes of IL-10 after one CHMI[17], which could serve to downmodulate the inflammatory response, but these responses have not been carefully tracked longitudinally in humans over defined repeat exposures to Pf-infected mosquitoes.

## Results

### Delayed patency

The clinical and immunologic responses of eight participants were followed over three to four exposures to mosquitoes infected with Pf strain NF54 parasites at intervals of 5–14 months (Fig. 1). Six of the eight participants were also exposed to uninfected mosquitoes as a mock challenge two months prior to their first *P. falciparum* exposure and five of these participants (all male) completed all four CHMI (Fig. 2). A positive ultrasensitive polymerase chain reaction (usPCR) preceded a positive thick blood smear (TBS) by an average of 3.55 days (range 1–8 days) and the average time to blood stage patency, defined as time from exposure to a positive TBS, lengthened with each CHMI (CHMI 1 11.46 ± 1 days, CHMI 2 12.22 ± 1.9 days, CHMI 3 13.25 ± 1.5 days, CHMI 4 13.8 ± 2.4 days, ANOVA test for trend $P = 0.007$) (Fig. 3A, B). The prepatent period among infectivity controls across all CHMI remained consistent with CHMI 1 results. The response varied among participants; five participants (2, 10, 16, 19, 27) had a delay in patency to

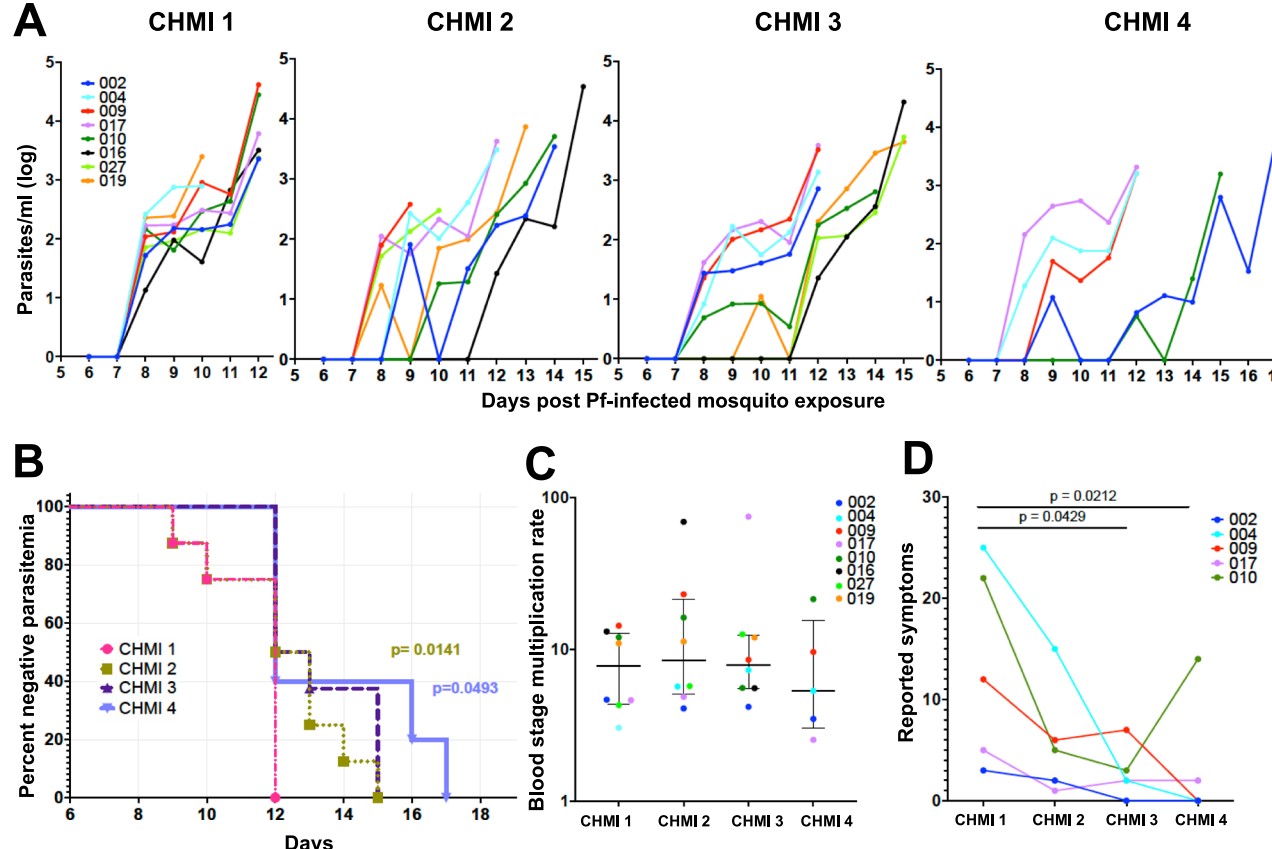

**Fig. 3 | Blood stage parasite detection was delayed and reports of symptoms decreased in CHMI 3 and 4. A** Parasitemia (p/ml) measured by quantitative PCR (qPCR) of Pf18s rRNA for each participant through the 4 CHMI (CHMI 1 n = 8, CHMI 2 n = 8, CHMI 3 n = 8, CHMI 4 n = 5)). Colored lines represent each participant. **B** Kaplan–Meier survival curve where each colored line indicates the percent of participants without patent parasitemia by blood smear on each CHMI. P value are based on the log-rank (Mantel–Cox) test comparing the time to first positive result between CHMI 1 and each other CHMI. **C** Blood stage multiplication was calculated for each participant (dots color coded as in (**A**)) for each CHMI using the increase in parasitemia over time from first qPCR detection until treatment. Data are presented as median and interquartile range. No significant differences were found between CHMI 1 and CHMI 2-4, Holm-Šídák's multiple comparisons test. **D** The total number of symptom reports by each participant that completed all 4 CHMI is indicated by a line (n = 5). Statistical differences were tested using the Dunn's multiple comparisons test comparing each CHMI to CHMI 1 and all P values are two-sided. Source data are provided as a Source Data file.

D15-17 in CHMI 3 or 4, while three (4, 9, 17) continued to become TBS positive by D12 even after CHMI 4. The average time to a positive usPCR also increased but was not significant (CHMI 1 8.09 ± 0.3 days, CHMI 2 8.89 ± 1.3 days, CHMI 3 9.38 ± 1.8 days, CHMI 4 9.2 ± 1.6 days). This modest change is likely due to the high sensitivity of usPCR and that only daily samples were collected, which limits our ability to observe parasite kinetics at a finer resolution. Once parasites appeared in the blood their replication rates were similar after each CHMI (Fig. 3C), suggesting the immune response affects pre-erythrocytic stages.

## Reduction in reported symptoms

Malaria-associated symptoms and fever declined with successive CHMIs. After CHMI 1, 100% of participants experienced headache, 75% malaise, 63% myalgia (1 severe), and 38% fever (2 severe, i.e., >39 °C) (Fig. 3D, Supplementary Fig 1). By contrast, no participants developed a fever in CHMI 4, and the only reported malaria symptoms were mild malaise in one participant and moderate myalgia, headache, and malaise in another, which was likely due to an unrelated viral infection. No serious adverse events occurred during the study and none of the clinical laboratory tests worsened progressively with repeat CHMI. No participants withdrew from the study due to intolerance to malaria symptoms. After each CHMI, AST and ALT, but not creatinine, levels were significantly higher at day of peak parasitemia (DRx) than baseline, but continued to be in the normal-mild elevation range in all

participants (Supplementary Fig 2). At DRx, there was also a trend toward decreased white blood cell (WBC) levels and hemoglobin (Hb) concentrations (Supplementary Fig 2) but these were not significant. WBC levels returned to baseline by D19-30 post-infection, while Hb levels did not return to baseline until the final follow-up visit on D41-54. A transient decline in platelets that largely remained within the normal range was noted. Again, all laboratory values post-CHMI were graded as normal or mild, except for one participant with a moderately low hemoglobin (11.9 & 11.5 g/dL) at DRx and D41, respectively during CHMI 4.

## Anti-CSP Ig levels correlate with time to patency

As an initial analysis of the immune response, antibody levels against the major sporozoite surface protein, PfCSP, and an antigen expressed by both liver and blood stages, glutamate rich protein (PfGLURP), were assessed using an enzyme-linked immunoassay (ELISA). Over the course of successive CHMIs, anti-PfCSP and/or PfGLURP antibodies gradually developed and were observed in all participants seven days after treatment (DRx+7) of their last CHMI (Fig. 4). The development of the response varied between participants, but there were common trends. Following each CHMI, anti-PfCSP and -PfGLURP antibody levels peaked on DRx+7, however, the increases were not significantly higher than baseline until CHMI 2. Antibody levels then declined back to the pre-CHMI baseline by D1 of the next CHMI. After CHMI 3 there was a more rapid increase in anti-PfCSP IgG and -PfGLURP IgG and IgM levels

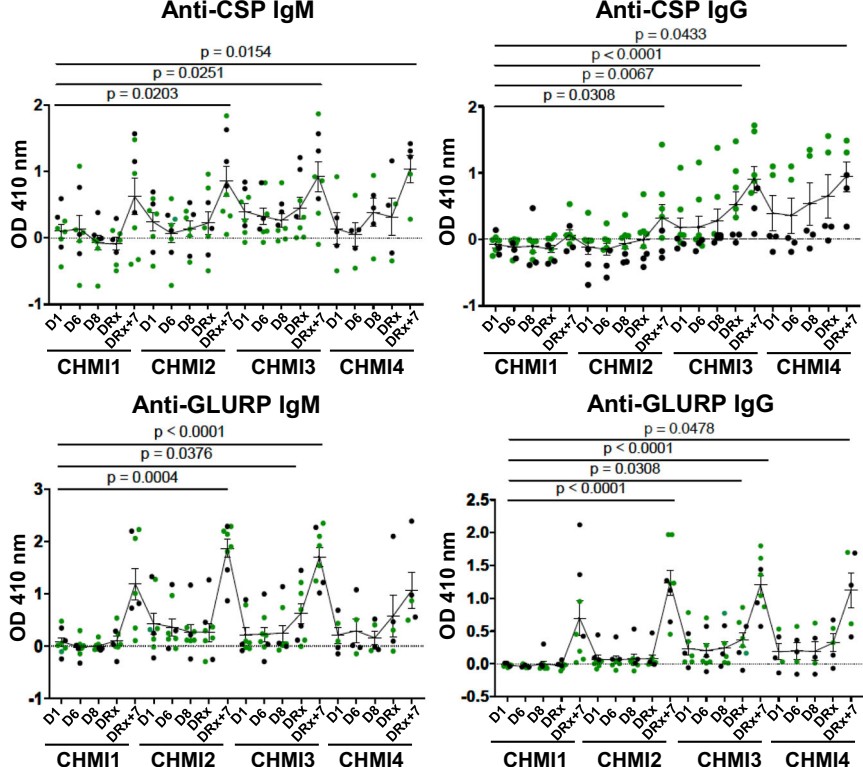

**Fig. 4 | Anti-CSP IgG levels increase after repetitive CHMI.** Plasma levels of IgM and IgG antibodies against CSP repeat (NANP$_8$) and the GLURP R2 repeat peptides were determined by ELISA for all participants (CHMI1 $n = 8$, CHMI2 $n = 8$, CHMI3 $n = 8$, CHMI4 $n = 5$) on D1 before exposure to Pf-infected mosquitoes, D6, D8, on day of treatment (DRx) and 7 days after treatment (DRx + 7) for each CHMI. For all participants the mean optical density (OD) at 410 nm at a 1/300 dilution subtracted by the OD 410 nm of a 1/300 dilution of the participants' baseline samples

(matched mock samples) is plotted. Participants with delayed patency at their last CHMI are shown in green. The population mean and standard error of the mean (SEM) at each time is indicated. Statistical differences were determined through CHMI 1-3 relative to D1 of CHMI 1, using Dunn's multiple comparisons test while due to the smaller samples size a Holm-Šídák's multiple comparisons test was used for CHMI 4. All $P$ values are two-sided and the source data are provided as a Source Data file.

that reached significance by DRx. Antibody levels continued to increase to DRx+7, but again by D1 of CHMI 4 mean anti-PfCSP and GLURP antibody levels had declined to baseline. Notably, the anti-PfCSP IgG levels for the two participants who had the longest patency delay remained elevated on D1 of CHMI 4 suggesting these two individuals had a more robust long-lived plasma cell response. Considering all participants across all the CHMIs, anti-PfCSP IgG levels on D1, DRx, and DRx+7 correlated significantly with time to patency ($P = 0.007$, 0.0001, and 0.0035, respectively) (Fig. 5A).

## B and T cell activation

The maturation of circulating T and B cell populations was also tracked by flow cytometry through the four CHMI on DRx +7 when Ig levels were highest. With each CHMI there was a trend toward increases in the percent of circulating IgG-positive plasmablasts, and atypical memory B cells as well as the number of activated memory B cells (Supplementary Fig. 3). Consistent with B cell activation there was also a significant increase in the number of activated CD4 + T cells (CCR7-CD45RO- and CCR7-CD45RO-CD69 + ) circulating on DRx+7 between CHMI 1 and 2 that remained high in later CHMIs (Supplementary Fig. 4). One participant, who was confirmed HIV-negative per study inclusion criteria, had extremely low numbers of CD4 + T cells ( ≤100 CD3 + CD4 + CD8- cells/200,000 PBMC), therefore they were excluded from the CD4 + T cell population percent calculations. The number of all CD3 + CD4 + T cells expressing the chemokine receptor CXCR6+ decreased significantly between CHMI 1 and 3 (Supplementary Fig. 4). Lower levels of the number and percent of CXCR6+ cells were also observed in all CD4+ populations from CHMI 2 on, including the CXCR6 + CCR7-CD45RO+ cells that also expressed the early activation

marker CD69+ and another chemokine receptor, CXCR3 + . Both CXCR6 and CXCR3 are associated with migration to non-lymphoid tissue, suggesting that these cell populations may be retained in tissues, such as the liver.

In contrast to CXCR6 + CD4 + T cells, the frequency of CD4 + T cells expressing CXCR5 increased significantly between CHMI 1 and 3 (Fig. 6A). This increase in the percent of CXCR5 + CD4 + T cells after the first CHMI was also significant in both effector (CCR7−) and central (CCR7 + ) memory (CD45RO + ) cell populations. CXCR5 along with CD45RO and PD-1 are markers for T follicular helper (Tfh) cells that play a key role in B cell maturation[22] and from CHMI 1 to 3 there was an increasing trend in this cell population, but it did not reach significance (Supplementary Fig 4). However, there was a positive correlation between each participant's Tfh cell number and their plasma CXCL13 level (Fig. 6B) and participants with higher anti-PfCSP IgG levels had significantly higher CXCL13 levels (Fig. 6C) consistent with enhanced B cell activation.

Decreases in CXCR6+ and increases in CXCR5+ populations were also observed for CD8 + T cells (Fig. 7 & Supplementary Fig 5). There was a significant decrease in the percent of CXCR6 + CCR7-CD45RO- cells between CHMI 1 and 3 and a significant increase in the number of CXCR5+ cells in both CCR7-CD45RO- and CCR7-CD45RO+ populations. In addition to CXCR5 + , the total number of CD8 + CCR7-CD45RO+ cells expressing chemokine receptor CXCR3+ or the inhibitory receptor PD-1[23] also significantly increased by CHMI 3 (Fig. 7B). Further analysis indicated that the number and percent of CXCR5 + PD1 + CD8 + T cells significantly increased by CHMI 2 and stayed elevated in CHMI 3 (Fig. 7D). This significant increase was observed in both CCR7-CD8+ populations (Fig. 7D). In contrast, only

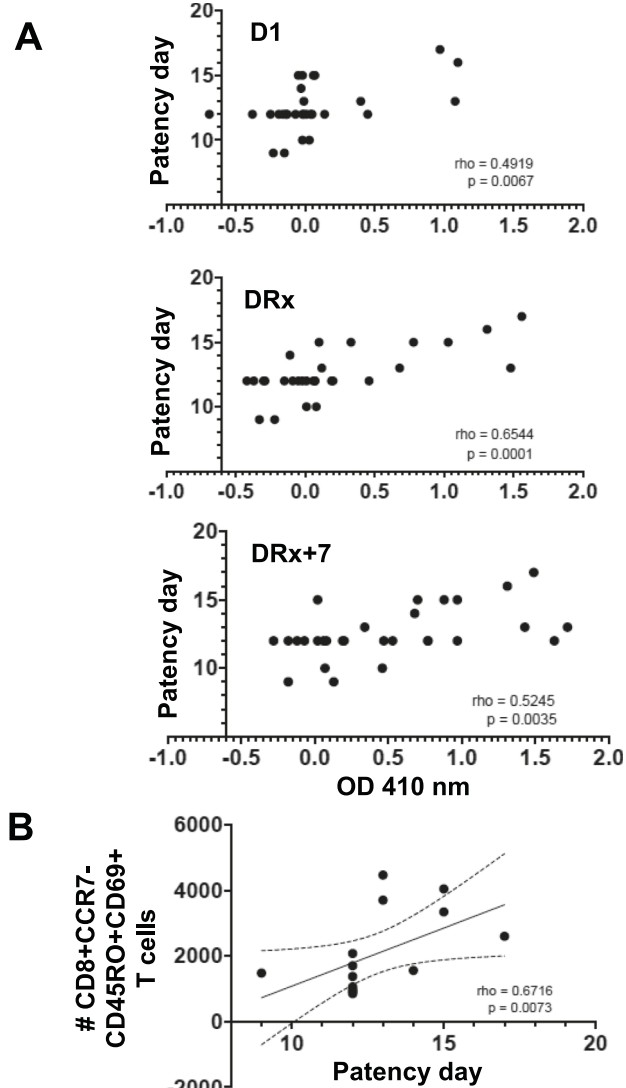

**Fig. 5 | Anti-CSP IgG levels and CD69 + CD8 + CCR7-CD45RO+ T cells are positively correlated with patency day. A** Patency day (number of days from mosquito exposure to a positive peripheral blood smear) for each CHMI participant is plotted against their anti-CSP IgG level determined by ELISA on D1, DRx or DRx +7 (CHMI 1 *n* = 8, CHMI 2 *n* = 8, CHMI 3 *n* = 8, CHMI 4 *n* = 5). **B** Patency day from CHMI 2 to CHMI 4 for each participant that completed all 4 CHMIs (*n* = 5) is plotted against the number of CD69 + CD8 + CCR7-CD45RO+ T cells detected using flow cytometry of unstimulated PBMC isolated after the same CHMI 7 days after treatment (DRx+7). All six participants that completed the mock CHMI were included in the flow cytometry analysis, (CHMI 1 *n* = 6, CHMI 2 *n* = 6, CHMI 3 *n* = 5, CHMI 4 *n* = 4). Statistical analysis was determined by Spearman correlation (GraphPad Prism 9). Rho = correlation coefficient and the two-sided *P* values (p) are shown. Source data are provided as a Source Data file.

the number and percent of PD-1 expressing cells significantly increased within the CCR7 + CD45RO + CD8+ population by CHMI 3 (Fig. 7C). Notably, the number of CD69 + CCR7-CD45RO + CD8 + T cells detected for each participant on DRx+7 after each CHMI was positively correlated with that participants' patency day for that particular CHMI (Fig. 5B), suggesting a role for CD8 + T cells in the control of pre-erythrocytic stage parasites.

**Pro- and anti-inflammatory cytokines increase**

In addition to antibodies, plasma was screened for 24 cytokines and all but four (TGFβ, IL-21, IFNα, and IFNβ) were detected in at least one sample including cytokines associated with B cell activation, BAFF,

APRIL and CXCL13 (Fig. 8A, Supplementary Fig 6). Increases over baseline were consistently observed only at DRx, except for CXCL10 and IL-18. CXCL10 levels peaked on DRx but were also elevated before (D6 and D8) and after DRx+7 (Fig. 8A, Supplementary Fig 7), while IL-18 levels increased by DRx and remained elevated through DRx+7 in most participants (Supplementary Fig 7). IL-10, CXCL10, and IL-1RA had the highest increase over baseline after each challenge. The levels of these three and all the other cytokines varied between individuals, with only CXCL10 and TNF consistently elevated above baseline in all participants after all four CHMIs at DRx (Supplementary Fig 7). IFNγ levels varied the most between individuals and between CHMIs (Supplementary Fig 7), yet were positively correlated with the most other cytokines (13 of the 20 cytokines detected) (Supplementary Fig 8). This result could suggest IFNγ plays a role in orchestrating the immune response to malaria infection. The levels of IFNγ and nine other cytokines also positively correlated with peak parasitemia and fever (Supplementary Fig 9). In an attempt to control for differences in parasitemia between participants and CHMI, each participant's DRx temperature and cytokine responses were compared for their first and second exposure to >3.3 parasites/μl (Fig. 8B). Greater than 3.3 parasites/μl was selected as the cut-off because it was the threshold for having a ratio over baseline >5 in any 2 cytokines in CHMI 1. There was a trend towards a decrease in temperature and the levels of the top five most upregulated cytokines during the participant's second exposure to high parasitemia, IL-10, CXCL10, IL-1RA, IFNγ, IL-6. However, only the decrease in IFNγ levels was significant.

## Discussion

The clinical and immunological responses of eight U.S. naïve participants to three or four sequential malaria exposures to the same strain of Pf by the bites of Pf-infected mosquitoes were monitored over the course of two years. After just two CHMIs, the time to bloodstream patency increased and the number of symptoms decreased, suggesting the development of parasitological and clinical immunity. Patency day was positively correlated with anti-PfCSP IgG levels on D1, DRx and DRx+7 as well as with the number of circulating CD8 + CCR7-CD45RO+ cells expressing the early activation marker, CD69, consistent with the gradual development of pre-erythrocytic immunity. The absence of a reduction in the growth rate of blood-stage parasitemia in CHMI 3 and 4 provided further support for the role of pre-erythrocytic immunity. Such partial pre-erythrocytic immunity is difficult to quantify in field studies[24], because it only reduces the number of merozoites released and extends the time required to detect blood-stage parasites. To measure such a delay in the appearance of blood stage parasites, the timing of Pf-infected mosquito exposure needs to be known. Our repeat CHMI model allows the direct evaluation of the development of the human immune response associated with this hidden and clinically silent part of the parasite life cycle.

Vaccine-induced pre-erythrocytic immunity has been reported after immunization with attenuated sporozoites[25,26] or sporozoite exposure with chloroquine or pyrimethamine prophylaxis to protect against blood stage parasites[27,28]. Immunity is dependent on sporozoite dose. Interestingly, when study participants taking chloroquine prophylaxis were exposed to five Pf-infected mosquitoes three times at monthly intervals five of the ten participants were protected against the appearance of any blood stage parasites following a homologous Pf sporozoite challenge 19 weeks later[29]. This result is different than the partial pre-erythrocytic immunity observed in our study, in which participants were exposed to five Pf infected mosquitoes three to four times at 6-9-month intervals and treated when parasitemia was detected by light microscopy. It is also distinct from the continued susceptibility of syphilis patients undergoing malaria fever treatment to rechallenge with Pf-infected mosquitoes[7]. The longer intervals between CHMIs could have reduced the immune response, as enhanced induction of CD8 + T cells has been demonstrated with

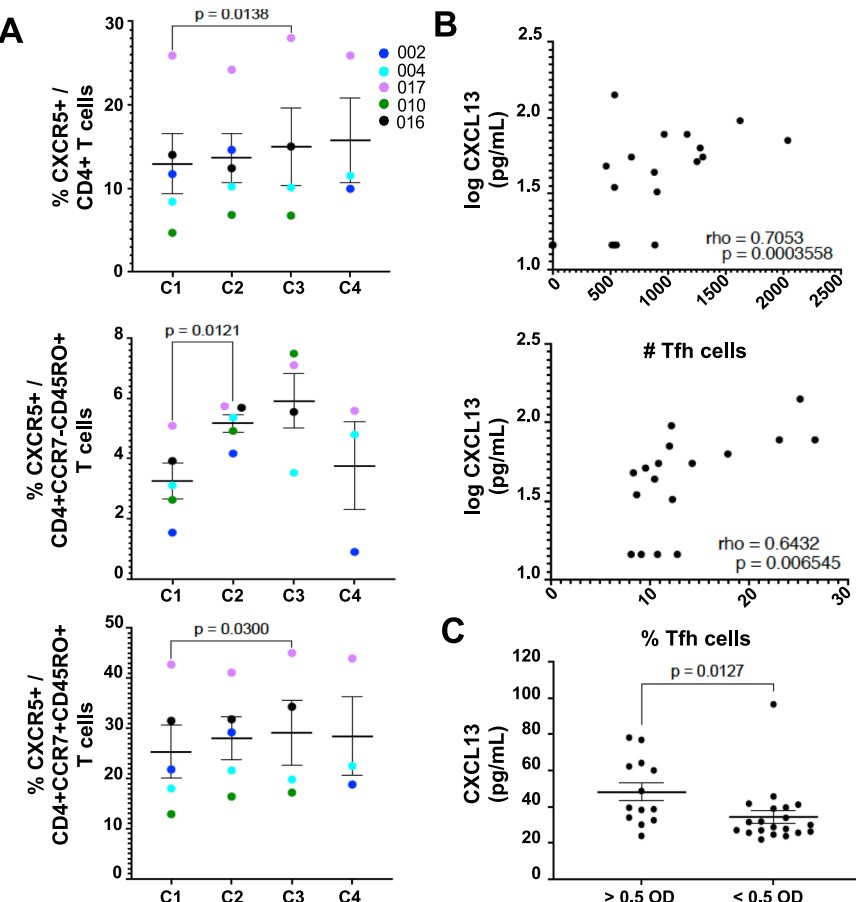

**Fig. 6 | CD4+CXCR5+ T cell populations increase after repetitive CHMI and T follicular helper cells (Tfh) correlate with CXCL13 levels.** PBMC from each participant collected 7 days after treatment (DRx+7) on each CHMI were analyzed by flow cytometry without exogenous stimulation. All six participants that completed the mock CHMI were included in the flow cytometry analysis (C1 $n = 6$, C2 $n = 6$, C3 $n = 5$, C4 $n = 4$), but the participant with a low CD3 + CD4 + T cell count was excluded from the percent calculations (C1 $n = 5$, C2 $n = 5$, C3 $n = 4$, C4 $n = 3$). **A** The percent of CXCR5 positive CD3 + CD4 + T cells, CD3 + CD4 + CCR7-CD45RO+ (effector memory) and CD3 + CD4 + CCR7 + CD45RO+ (central memory) T cells detected on DRx+7 of each CHMI is shown. Mean values +/− standard error of the mean (SEM) are indicated for each time point. Statistical differences were determined using Dunnett's multiple comparisons test between CHMI 1 and CHMI 2-4.

Data that did not pass the normality test was first logarithmically transformed to conform to normality and then tested for significance. All *P* values are two-sided. **B** The logarithm of plasma CXCL13 levels (pg/ml) on DRx for each participant for each CHMI is plotted against the number (upper graph) or percent (lower graph) of (CD3 + CD4 + CD45RO + CXCR5 + PD-1+ (Tfh)) cells detected in the DRx+7 PBMC sample from the corresponding participant at the corresponding CHMI. Correlation analysis was determined using Spearman's rank correlation and listed in each panel (rho = correlation coefficient and *p* = *P* value). **C** The mean value +/− SEM of CXCL13 (pg/ml) in plasma with anti-CSP IgG levels above or below 0.5 OD 410 nm on D1, D6, D8, DRx and DRx+7 for all 4 CHMI is shown. Statistical differences were determined using Mann–Whitney test and all *P* values are two-sided. Source data are provided as a Source Data file.

---

multidose delivery of the Ty21A *Salmonella* Typhi vaccine[30,31] and vaccine efficacy improved with multi-dose primed PfSPZ Vaccine delivery[32], presumed due to CD8 + T cell recruitment[33]. However, it is also intriguing to speculate that the expansion of blood stage parasites in our clinical trial could inhibit the generation of a strong pre-erythrocytic immune response. In murine malaria models such an inhibitory role of blood stage parasites on pre-erythrocytic immunity has been reported[21].

Both PfCSP antibodies and T cells have been associated with vaccine induced pre-erythrocytic infection[2] and treatment with PfCSP-specific monoclonal antibodies have been shown to block liver stage infection in humans[34]. PfCSP-specific vaccines, RTS,S[35] and R21[36], are also partially effective but do not confer sterilizing protection in the majority of recipients. In mice, CD8+ liver-resident memory T cells (Trm) are required for the protective immune response generated against radiation-attenuated sporozoite vaccines[37]. Murine Trm associated with malaria protection express CD69+ and CXCR6 + [37], which are also expressed by Trm cells in humans[38]. Due to the risks involved in isolating liver samples from humans, our analysis was limited to peripheral blood, which has been shown to contain T cells with an expression profile similar to Trm cells identified in liver biopsies[38]. These circulating cells could be Trm precursors or a subset of Trm that have been transiently released into the peripheral blood before returning to the liver. By CHMI 3, the significant decreases in the percent of circulating CXCR6 + CCR7-CD45RO-CD8+ cells and the total number of CD4 + T cells expressing CXCR6, as well as the decreasing trend of CXCR6+ expression in the other CD4+ cell populations is consistent with both CXCR6 + CD8+ and CD4 + T cell retention in the liver, but this needs to be further investigated.

Conversely, CXCR5 + CD4+ and CXCR5 + CD8+ cell populations increased after CHMI 1. CXCR5 mediates trafficking to B cell-rich regions suggesting that these CXCR5+ cell populations could co-localize in germinal center follicles[39]. The role of CXCR5+ Tfh cells in B cell activation is well established and CXCR5 + CCR7- CD8 + T cells have been reported to localize in lymphoid tissue. However, the role of CXCR5 + CD8+ in B cell activation is more speculative[40]. CXCR5 + CD8 + T cells that co-express PD-1+ and the transcription factor, Tcf1, have been shown to be induced in response to chronic viral infections and tumors and expansion after treatment with PD-1 blockers is associated with clearance[39,41]. This is the first report of an

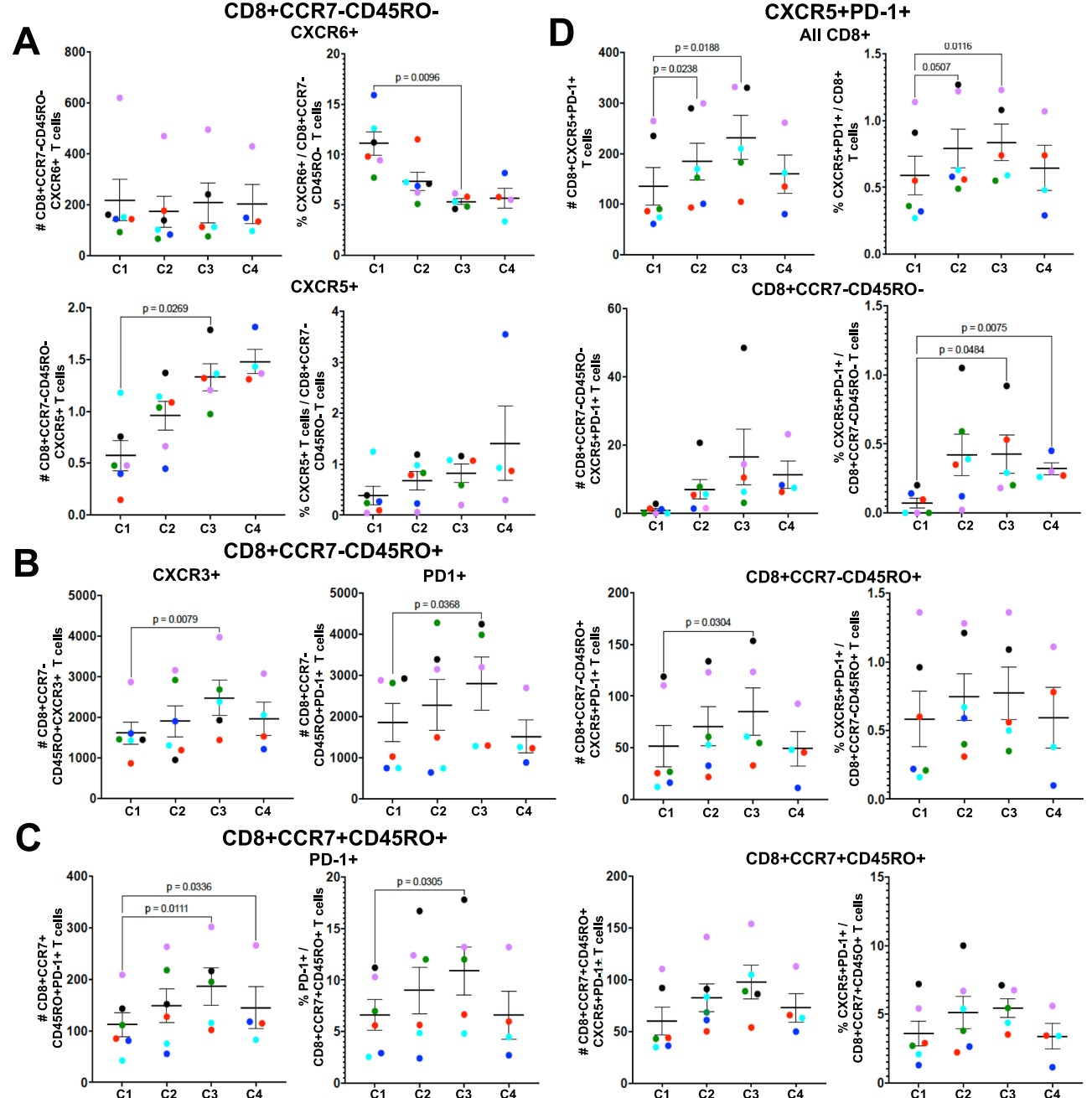

**Fig. 7 | CD8 + CCR7-CD45RO-CXCR6 + T cells decrease and CD8 + CCR7-CXCR5 + T cells increase after repetitive CHMI.** Flow cytometry was used to analyze PBMC collected 7 days after treatment (DRx+7) from all participants that completed the mock CHMI analysis (C1 n = 6, C2 n = 6, C3 n = 5, C4 n = 4). **A** The number of cells per 200,000 total cells and percent of CD3 + CD8 + CCR7-CD45RO- (effector) T cells positive for CXCR6 or CXCR5. **B** The number of CD3 + CD8 + CCR7-CD45RO+ (effector memory) cells per 200,000 total cells positive for CXCR3

or PD-1. **C** The number and percent of PD-1 + CD8 + CCR7 + CD45RO+ (central memory) T cells. **D** The number and percent of CXCR5 and PD-1 positive CD8 + T cells, CD8 + CCR7-CD45RO−, CD8 + CCR7-CD45RO+ and CD8 + CCR7 + CD45RO + T cells. Data are presented as mean values +/− standard error of the mean (SEM). Statistical differences were determined using Dunnett's multiple comparisons test between CHMI 1 and CHMI 2-4 and all *P* values are two-sided. Source data are provided as a Source Data file.

increase in CXCR5 + PD-1 + CCR7-CD8+ cells following sequential Pf infections and further research is needed into better understand their phenotype and function. There was also an increase in PD-1 + CD45RO + CCR7 + CD8+ cells by CHMI 3 consistent with changes observed in response to repetitive challenges with other pathogens[42–44] that could modulate the response of this central memory cell population to future infections. The expansion of these T cell populations could contribute to the decline in the response observed after CHMI 4 as well as the lack of sterilizing immunity against *Plasmodium*. Further functional evaluation of the CD8+ cell

response is required to direct the development of vaccine strategies to enhance parasite-specific, liver-resident CD8+ and CD4 + T cells in addition to antibody responses.

Additional support for the role of pre-erythrocytic immunity in delaying the onset of patency is suggested by an alternate CHMI model that challenged participants with blood-stage parasites and did not observe a patency delay[45,46]. Conversely, at peak parasitemia on treatment day the cytokine profile for both studies is similar, including an increase in both IL-10 and IL-1RA levels that could reduce inflammation[45]. This finding also suggests that blood stage, not

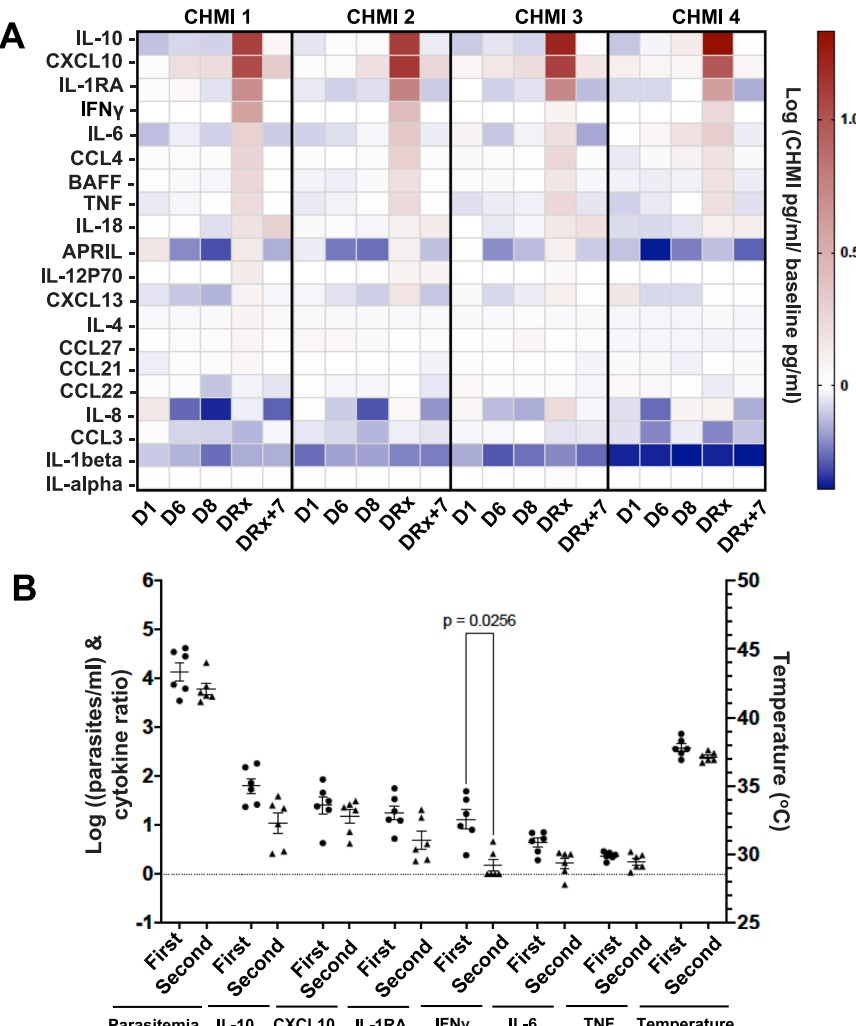

**Fig. 8 | Cytokines levels peaked with parasitemia, except IL-18 that continued to increase for seven days.** Plasma cytokine levels were measured by multiplex bead analysis for each participant (CHMI 1 $n$ = 8, CHMI 2 $n$ = 8, CHMI 3 $n$ = 8, CHMI 4 $n$ = 5) on D1 before exposure to Pf-infected mosquitoes, D6, D8, day of treatment (DRx) and 7 days after treatment (DRx+7) for each CHMI. **A** The heatmap of the log of the average ratio of the cytokine concentration (pg/ml) on the indicated day of the CHMI to the baseline concentration following the mock challenge for all participants is shown. **B** Comparison of the parasitemia (parasites/ml) measured by quantitative PCR (qPCR) of Pf18s rRNA, the ratio of the plasma cytokine levels to the mock CHMI baseline quantified using a multiplex bead assay, and maximal temperature (°C) measured on the day each participant experienced their first and second peak of >3.3 parasites/µl ($n$ = 6). Data are presented as mean values +/− standard error of the mean (SEM). Statistical differences were tested using Sidak's multiple comparisons test and the $P$ value is two-sided. Source data are provided as a Source Data file.

pre-erythrocytic, parasites are the major stimulus for cytokine secretion. The blood stage challenge also induced a strong innate and adaptive T cell response that decreased after repeat exposures[45]. Most notably, after repeated blood stage challenges, the CD4 + T cell response was less diverse and lower in magnitude. We did not see evidence of an overall decrease in CD4+ populations in our longitudinal repeat Pf-infected mosquito CHMIs, but as previously reported there was a marked increase in γδ2 T cells after the initial two exposures that declined in later challenges, while the number of γδ1 T cells increased in later CHMIs[15].

Both studies are limited by the small number of participants and the interruption in immunologic response necessitated by the need to treat individuals at an early stage at first detection on a thick smear blood smear. This early treatment may delay or reduce the immunologic response against blood stage parasites previously reported during the prolonged high parasitemias used for malaria therapy treatment of syphilis[6,8]. Unlike our longitudinal study, the repeat blood stage challenge was a cross-sectional study analyzing distinct sets of participants after different numbers of challenges. Our use of

Pf-infected mosquitoes mimics natural exposure to both the pre-erythrocytic and erythrocytic stages of the life cycle, but it is more difficult to control the parasite burden between participants and CHMIs. Due to the vicissitudes of participation over multiple years, attrition resulted in an all-male panel of participants at study end which may bias study results. The separation of the repeat CHMIs by intervals of greater than 6 months could also influence the response.

The stimulation of both anti- and pro-inflammatory cytokines following CHMI is consistent with data following a single CHMI[17], field studies of naturally acquired infections[47–49], and the repeat blood stage infection study reported above[45]. Neither in our study nor these other studies of uncomplicated malaria were IL-1β levels increased in plasma. IL-1β levels do increase in severe malaria and could require a higher level of antigen or inflammation to stimulate cleavage and secretion of the inactive pro-peptide[47,50].

In light of the decrease in symptoms observed after CHMIs 3 and 4, we anticipated a decrease in the pro-inflammatory cytokines and an increase in the inhibitory cytokines in response to a second exposure to high parasitemia. However, of all the circulating cytokines tested

only IFNγ levels were significantly lower during the second high parasite exposure and none trended higher, including IL-10 and IL-1RA. It is possible that changes in the γδ1 and γδ2 T cell populations through repetitive CHMIs[15] could contribute to the reduction in IFNγ secretion and modulate monocyte activation leading to fewer symptoms. There could also be local changes in cytokine levels in tissues that were not assessed in this study. Changes in p53 expression in monocytes isolated from immune individuals in the field have been associated with a reduction in symptoms[12] and could be regulated by epigenetic changes. The continued increase in IL-18 levels for a week after treatment may also play a role modulating the response. A similar increase in IL-18 was observed 6 days post-treatment in the repeat blood stage infection study and its expression pattern correlated with activated CD4+ and activated regulatory T cells[45]. In field studies, plasma IL-18 levels increase during uncomplicated and severe clinical malaria cases and remain elevated at follow-up[51]. In vitro, in the presence of IL-12, IL-18 stimulates differentiation of T helper 1 (Th1) cells and also stimulates NK, B, and γδ and Th1 cells to produce IFNγ[52] and IL-18 has a protective role in rodent malaria by increasing NK cell IFNγ production[53]. The expression of IL-18 receptors on both T cells and B cells suggest it could continue to modulate maturation of the adaptive response, possibly interfering with memory cell production.

In summary, this work demonstrates that as few as two exposures to the same Pf strain in the absence of prophylactic chemotherapy can induce a protective but not sterilizing immune response against pre-erythrocytic stage parasites, including anti-PfCSP IgG levels and an increase in activated CD8+ effector memory cells that correlate with the increased time to patency measured by microscopy. The concomitant progressive decrease in the circulating CXCR6 + CD8+ and CD4 + T cell populations after CHMI 1 is also consistent with enhanced retention in the liver and more effective parasite clearance. While the increase in PD-1 + CD8+ central and effector memory cells after CHMI 3 could reduce subsequent responses and interfere with the generation of robust sterilizing immunity. These findings support the development of vaccine strategies that modulate the CD8+ response to more effectively control the parasite. In contrast to the delayed patency, even after four exposures to the same parasite strain, there was no evidence of an effective response against blood-stage parasites and 7 days after peak parasitemia there was only a gradual increase in the number and percent of the circulating activated, classical, and atypical memory B cells and Tfh cells that never reached significance. This slow acquisition of immunity is similar to that seen in the field and suggests it is an intrinsic property of the parasite-host interaction, not solely due to strain polymorphisms. Additional evaluation of the response of malaria-specific memory B and T cell populations isolated after each challenge to antigen and specific cytokines, along with transcriptomic and epigenetic testing, will provide insight into the factors that influence memory cell production and inform future vaccine development.

## Methods

### Controlled Human Malaria Infection (CHMI)
A prospective Phase 1 cohort study was conducted to assess the initial induction and subsequent maturation of the immune response during sequential exposure to *P. falciparum* infected mosquitoes over 2–3 years (ClinicalTrials.gov# NCT03014258). CHMI were conducted at the University of Maryland, Baltimore (UMB) Center for Vaccine Development and Global Health using *Anopheles stephensi* mosquitoes infected with *P. falciparum* strain NF54 sporozoites produced and assessed on-site by Sanaria Inc. (Rockville, MD). Healthy, malaria-naïve, male and female U.S. participants ages 18–50 years, inclusive, from the greater Baltimore–Washington area, were recruited by UMB. All participants provided informed consent prior to study activities. Eligibility criteria for participant enrollment and the Study Protocol are available in Supplementary Material (pgs 15–126). Criteria were assessed by self-reported medical history, clinical laboratory tests, and physical examination. Briefly, the participants had to be non-pregnant and non-lactating, have a body mass index $\leq$35 kg/m$^2$ and a negative sickle cell screening test. Laboratory studies within 56 days prior to enrollment had to demonstrate normal hemoglobin, platelet count, alanine aminotransferase and serum creatinine and negative serologies for HIV, hepatitis B, and hepatitis C. Absence of pregnancy was assessed throughout the study and participants had to agree not to travel to a malaria endemic region during the entire course of the study. Exclusion criteria included history of malaria infection, receipt of a malaria vaccine, splenectomy, sickle cell disease, sickle cell trait or adverse response to mosquito-bites or malaria medications.

Enrolled participants were first exposed to the bites of 5 uninfected (mock CHMI) *Anopheles stephensi* mosquito. Subsequently they underwent 3 or 4 infective challenges (CHMI 1-4) at months 2, 9, 14 and 23. A complete CHMI for each subject consisted of bites by five mosquitoes documented to have taken a blood meal and to be infected with *P. falciparum* sporozoites with a salivary gland score of 2 or greater (100–1000) following standard procedure[54]. For the final three CHMIs (CHMI #s 2-4), five to six additional malaria-naïve participants were enrolled and challenged as infectivity controls. To replace dropouts, two of the infectivity controls from CHMI 2 were recruited to undergo successive CHMI with the core group (Fig. 2). Solicited and unsolicited adverse events were monitored from days 6-28. Evaluations for parasitemia by ultrasensitive polymerase chain reaction (usPCR)[55] used primers 5' CCGACTAGGTGTTGGATGAAAGTGTTAA 3' and 5'AACCCAAAGACTTTGATTTCTCATAA 3' with the fluorescently tagged probe 5' Cy5-AGCAATCTAAAAGTCACCTCGAAAGATGACT-BHQ2 3' and after an initial incubation for 20 m at 50 °C followed by a 15 m incubation at 95 °C the reaction was cycled 45 times between 94 °C for 45 s and 60° for 75 s with a 1.7 °C/s ramp time. Thick blood smears (TBS) began on post-CHMI days 5 and 8 respectively and continued until parasite clearance was confirmed on TBS. Treatment occurred following TBS positivity, defined as two unquestionable parasites, confirmed by two investigators after five passes (0.5 μL) on the short axis (increased to 10 passes if symptomatic) of a 1 × 2 cm thick smear with 10 μL blood. Participants were provided directly observed Malarone® (Atovaquone/proguanil), or Coartem® (artemether/lumefantrine) as secondary therapy, and completed outpatient follow-up through day 56.

The sponsor was the National Institute of Allergy and Infectious Diseases (NIAID), at the National Institutes of Health (NIH) in Bethesda, MD, USA. The clinical trial protocol was reviewed and approved by the UMB Institutional Review Board (IRB) through a reliance agreement established with the USU IRB. After the fourth CHMI, the protocol was amended to include a fifth CHMI that was scheduled to occur in April 2020 and the participants were reconsented, but the CHMI was canceled due to COVID-19 restrictions.

### Serology
*Enzyme-Linked Immunoassay (ELISA)* To evaluate antibody levels, ELISAs were performed using plasma samples collected on D1 before exposure to Pf-infected mosquitoes, D6, D8, day of treatment (DRx) and 7 days after treatment (DRx + 7) for each CHMI. Briefly, Nunc Maxisorp (Invitrogen) 96 well plates were coated separately with 2 synthetic peptides (GenScript): the CSP repeat, amino acid sequence (NANP)$_8$ or the GLURP R2 repeat region, amino acid sequence (CGDKNEKGQHEIVEVEEILPEGC). Peptides diluted to 2 ug/ul with 1X phosphate buffered saline (PBS, pH 7.2) (KD Medical) were incubated overnight at room temperature (RT). Plates were washed 4 times with 0.5% Tween 20 (KPL) in 1X PBS and then blocked for 1 h at RT in blocking buffer (1x PBS with 5% BSA (Sigma Aldrich) and 0.05% Tween 20) before adding serially diluted samples from CHMI participants or, as negative controls, serum from malaria naïve donors (Interstate Blood Bank). Positive controls included mouse monoclonal antibody anti-PfCSP (1/10000 dilution, BEI resources) or pooled serum from

malaria patients (1/100 dilution). After incubation for 2 h at RT, the plate was washed 4 times and the appropriate horseradish peroxidase (HRP)-labeled secondary antibodies were added: goat anti-human IgG (1/1000 dilution, Life Technologies), goat anti-human IgM (1/2000 dilution, Invitrogen) and incubated at RT for 1 h. After 4 washes, 2,2′-Azino-bis(3-ethylbenzothiazoline-6-sulfonic acid) peroxidase substrate (KPL International Ltd) was added and after 1 h at RT the reaction was stopped with 1% SDS (Fisher Scientific) and absorbance read 410 nm using a VersaMax Tunable Microplate Reader (Molecular Devices). All samples were tested in duplicate. Antibody reactivity was defined as the optical density at 410 nm (OD410) of a 1/300 dilution of the subject's plasma sample subtracted by the OD410 of a 1/300 dilution of the subject's baseline sample (matched mock samples). For 6 of the 8 subjects, baseline values were the calculated as the average of the plasma samples ODs obtained during the uninfectious challenge (mock). For the remaining 2 subjects, baseline levels were based on the OD of the only sample obtained before the subject's first mosquito infectious challenge.

*Magnetic Luminex Performance Assay* Plasma cytokines were analyzed for each volunteer on D1 before exposure to Pf-infected mosquitoes, D6, D8, day of treatment (DRx) and 7 days after treatment (DRx + 7) for each CHMI. IFNγ, IL-10, IL-6, TNF alpha, IL-8, IL-12, IL-1 beta, IL-4 levels were measured using the human high sensitivity cytokine premixed bead kit A (R&D Systems), while standard premixed bead kits (R&D Systems) were used for the rest. All samples were centrifuged at $16,000 \times g$ for 4 min before use and diluted 2-fold with calibrator diluent RD6-40. Seven standards were serially diluted 4-fold with calibrator diluent RD6-40. The microparticle suspension was diluted 1:2 in microparticle diluent and the biotin-antibody cocktail vial was diluted 1:4.5 in biotin antibody diluent 2 as indicated by the manufacturer.

One hundred microliters of samples or standards and 25 ul of the microparticle cocktail were added per well and after sealing they were incubated for 3 h at RT on a horizontal orbital microplate shaker (0.12″ orbit) at $800 \pm 50$ rpm. Plates were then washed 3 times using a handheld magnetic plate washer (ThermoFisher Scientific) before 50 ul of diluted biotin-antibody cocktail was added to each well. Plates were sealed, incubated for 1 h at RT on the shaker ($800 \pm 50$ rpm), washed and then 50 ul of streptavidin-PE were added to each well. After sealing, incubation for 30 min RT on the shaker ($800 \pm 50$ rpm) and washing, the microparticles were resuspended in 100 μl of wash buffer, incubated for 2 m on the shaker ($800 \pm 50$ rpm) and read on a Bio-Rad analyzer within 90 m of completing the assay.

BAFF, CCL4, April, IL-1 alpha, IL-21, IFN beta, IFN alpha, IL-18, IL-1RA, CXCL10, CCL22, CXCL13, CCL27, CCL21, CCL3 levels were measured using the human pre-mixed multi-analyte bead kit (R&D Systems). All samples were centrifuged at $16,000 \times g$ for 4 min before use then diluted 2-fold with calibrator diluent RD6-52. Six standards were serially diluted 3-fold with calibrator diluent RD6-52. The microparticle and biotin-antibody cocktails were diluted 10x in diluent RD2-1 and the streptavidin-PE concentrate was diluted 25-fold as indicated by the manufacturer. The rest of the procedures were the same as described above for the human high sensitivity cytokine premixed kit A.

TGF beta was quantified using the human TGF-B1 kit DuoSet ELISA (R&D Systems) according to the manufacturer's instructions. Briefly, 96 well plates containing 100 ul of diluted capture antibody per well were sealed and incubated overnight at RT. Plates were washed 3 times, then blocked for 1 h at RT before 3 more washes. One hundred microliters of diluted samples (2-fold dilution) and 7 standards (2-fold serial dilutions) were added per well. After incubation for 2 h at RT the washing step was repeated and 100 μl of a working solution of Streptavidin-HRP B were added per well. Plates were incubated for 20 m at RT, washed 3 times and then 100 ul of substrate solution were added to each well and incubated for 20 m at RT. Fifty microliters of stop solution were then added and absorbance at 450 nm was read immediately.

For analysis, the concentration of each cytokine at each time point was determined based on the standard curve. Values below the observed concentration of the lowest valid standard (OOR <) were replaced by the values of their corresponding lowest standard. Values above the observed concentration of the lowest valid standard (OOR >) were replaced by their corresponding highest standard. To compare different cytokines, we used the ratio of the cytokine concentration on each day (CHMI pg/ml) to the baseline cytokine pg/ml (the average concentration obtained during the mock challenge) using the formula [cytokine concentration (pg/ml) / average cytokine concentration (pg/ml) during the mock challenge] = Cytokine day/baseline ratio.

## Cellular immunity

*Peripheral Blood Mononuclear Cells (PBMC) isolation* For the phenotypic determination of the lymphocyte populations, PBMCs were isolated by apheresis 7 days after treatment (DRx + 7) for each CHMI. Briefly, blood cells were resuspended in 1x phosphate-buffered saline (PBS, pH 7.2) (KD Medical) and 30 ml of diluted cells were layered on 20 ml of Ficoll-Paque Plus (Ficoll – GE Healthcare) in a 50 ml Leucosep TM tube (Greiner Bio One) that had been prepared by centrifugation at $1000 \times g$ for 1 m at RT with acceleration set to 3 and no brake. After adding the cells, the tubes were centrifuged for 10 m at $1000 \times g$ at RT with acceleration 3 and no brake. The supernatants were removed and the lymphocytes/PBMC fraction was collected, diluted with 15 ml of 1X PBS and centrifuged for 10 m at $200 \times g$, acceleration 9 and deacceleration 9, at 15 °C. The pellets were resuspended with 10 ml 1X PBS and cell viability assessed using a hemocytometer. The cells were then pelleted by centrifugation at 200 g for 10 m at 4 °C (acceleration 9, deceleration 9), and resuspended at $1 \times 10^7$ cells/ml in freezing media (RPMI (KD Medical), FBS (50%, Atlanta Biologicals), L-Glutamine (2 mM, Invitrogen), gentamycin (50 g/ml, Gibco), sodium pyruvate (2.5 mM, Gibco) and HEPES (10 mM, Gibco)) with 10% DMSO. Cells were aliquoted to cryovials, frozen overnight at −80 in Mr. Frosty freezing containers (Nalgene) and the following day stored in liquid nitrogen.

*Cell immunophenotyping by flow cytometry* Multi-parameter flow cytometry was used to assess unstimulated T and B cells from peripheral blood mononuclear cells (PBMC) isolated 7 days after treatment (DRx + 7) for each CHMI, as described above. For use in this assay, cryovials of PBMC samples were thawed for 60 s at 37 °C. One milliliter of warm 1:19 diluted BSA stain buffer was added dropwise at a rate of 1 ml/5 s and mixed by inversion twice. Samples were transferred to 5 ml of warm BSA stain buffer and centrifuged at 3000 g for 5 m (brake 2) before washing with 3 ml of BSA stain buffer and centrifugation at $300 \times g$ for 5 m (brake 2). Cells were then resuspended in 1 ml of BSA stain buffer, counted, and $1 \times 10^6$ cells were placed in each flow tube.

For the Live/Dead stain tube, cells were washed once with 1 ml of 1X PBS (pH 7.2, KD Medical), centrifuged at 3000 g for 5 m (brake 2) and the cell pellet resuspended in 1 ml 1X PBS and 1 ul reconstituted fluorescent reactive dye. After incubation on ice for 30 m protected from light, the cells were washed with BSA stain buffer (3 ml), centrifuged at $3000 \times g$ for 5 m (brake 2) and then resuspended in BSA stain buffer (100 μl). For the B cell panel, non-specific Fc-mediated interactions were inhibited by adding 2.5 μg of Fc block (BD Biosciences) per tube of $1 \times 10^6$ PBMC and incubating at RT for 10 m. The following fluorescent antibodies (all from BD Biosciences) were added and mixed well: 20 μl CD21-PE, 5 μl CD24-PE-CF594, 5 μl CD19-APC, 5 μl IgG-APC-Cy7, 5 μl CD27-BV421, 5 μl CD10-BV605, 5 μl CD38-BV711 and 5 μl CD20-BUV395. For the T cell panel, the following fluorescent antibodies were added and mixed well: 20 μl CD3-FITC (BD Biosciences), 5 μl CXCR6-PE (Biolegend), 5 μl CD69-PE-CF594 (BD Biosciences), 20 μl PD-1-APC (BD Biosciences), 5 μl CD8-APC-Cy7 (BD Biosciences), 5 μl CD45RO-BV421 (BD Biosciences), 5 μl CD4-BV510 (BD

Biosciences), 5 µl CXCR5-BV605 (Biolegend), 5 µl CCR7-BV650 (Biolegend) and 5 µl CXCR3-BV711 (BD Biosciences). All flow tubes were incubated at RT for 25 m protected from light, then washed with BSA stain buffer (3 ml) and centrifuged at 3000 × *g* for 5 m (brake 2). The cell pellet was incubated in 500 ul of Biolegend Fluorofix Buffer at RT for 30 m, then washed with BSA stain buffer (3 ml) and resuspended in 500 ul of BSA stain buffer. All tubes were analyzed in the LSRII cytometer (BD Biosciences) the following day. Data was analyzed using FlowJo software version 10.8.2 and the gating strategies are shown in Supplementary Fig 10.

## Statistical analysis

For determination of the significance in patency delay between CHMIs, the time to first positive result between CHMI 1 and each other infection was examined (Mantel−Cox test). To determine blood stage multiplication rates (BSMR) between CHMI, a log linear regression for each patient of the log (iRBC) numbers as a function of day was calculated. The formula $BSMR = 10^{2m}$, where $\log [iRBC(t)] = mt + b$ and $t =$ time (x axis), $m =$ slope and $b =$ intercept with Y axis was used. To assess the differences between BSMR and CHMI, a Holm-Šídák's multiple comparisons test was performed. For comparison of the total number of reported symptoms at each CHMI, a Dunn's multiple comparison test was used. To evaluate the differences between the laboratory clinical parameters and the CHMIs and to determine B and T cell population difference in individuals between CHMIs, a Dunnett's multiple comparisons test was performed. In the cases where data did not pass the normality test, it was first logarithmically transformed. The variation in CXCL13 concentrations at high and low antibody levels was analyzed using a Mann−Whitney test. To assess differences between cytokines, parasitemia and temperature according to parasitemia peaks, a Sidak's multiple comparison test was performed. For all correlation analysis (antibody levels vs time to patency, cytokine levels vs percentage of cells, cytokine levels vs cytokines levels, cytokine levels vs parasitemia, cytokine levels vs temperature and numbers of CD8 + CD69+ effector memory cells vs time to patency), a Spearman correlation analysis was done. All analyses were performed using GraphPad Prism version 9·5·1. Statistical significance was considered below 0.05 and a corresponding confidence level of 95%.

## Reporting summary

Further information on research design is available in the Nature Portfolio Reporting Summary linked to this article.

## Data availability

De-identified data reporting the study's primary and secondary protocol-specified objectives are shown in Figs. 3 and 4 and the source data for all the figures are provided as a Source Data file. The study protocol and inclusion exclusion criteria are included in the Supplementary Information, pgs 15–126 and available on ClinicalTrials.gov # NCT03014258. For investigators whose proposed use of the data are approved by the UMB IRB, additional de-identified data will be made available within a month following a request to the corresponding authors. Source data are provided with this paper.

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

## Acknowledgements

We thank the clinical trial participants for their contributions and commitment to malaria research. We additionally thank the sponsor, NIAID and Emmes Corporation for clinical trial support, University of Maryland Center for Vaccine Development and Global Health for regulatory support and Kateryna Lund at the Uniformed Services University of the Health Sciences Biomedical Instrumentation Center for assistance with flow cytometry. The Sponsor assisted with protocol development and safety monitoring, not study design, data collection and analysis, or manuscript writing. This work was supported by grants from the National Institute of Allergy and Infectious Diseases R01AI110852 (KCW), U01AI110852 (KCW) and 1R01AI177698 (KCW).

## Author contributions

K.C.W. was the principal investigator and K.E.L. was clinical site principal investigator for the study. K.C.W., K.E.L., S.L.H., E.N., G.A.D. and A.A.B. contributed to the conception and design. K.E.L., A.A.B., F.P.B., S.L.H., B.K.L.S., P.F.B., Y.A., A.P., S.C., B.S., K.S., P.F., K.K., M.B., S.M.G., M.U., S.K.P. and D.M.R. contributed to investigation and sample collection. K.C.W., K.E.L., P.F., A.N.B., A.A.B., G.A.D. and S.K.P. contributed to data analysis and interpretation. K.E.L., K.C.W., A.A.B. and P.F. have accessed and verified all the underlying data. All authors contributed to the writing and final approval of the manuscript.

## Competing interests

The authors declare no completing interests. Material has been reviewed by the University of Maryland Institutional Review Board. There is no objection to its presentation and/or publication. The opinions or assertions expressed are the private views of the authors, and are not to be construed as official, or as reflecting true views of the Uniformed Services University of the Health Sciences, Henry M. Jackson Foundation for the Advancement of Military Medicine, Inc. or the Department of Defense. The investigators have adhered to the policies for protection of human subjects as prescribed in AR 70–25. This work was supported by NIAID of the National Institutes of Health (NIH) under award numbers R01AI110852 and U01AI110852; Clinical Trials.gov registration number, NCT03014258.

## Additional information

[1]Department of Microbiology and Immunology, Uniformed Services University of the Health Sciences, Bethesda, MD, USA. [2]Henry M. Jackson Foundation for the Advancement of Military Medicine, Inc., Rockville, MD, USA. [3]Center for Vaccine Development and Global Health, University of Maryland School of Medicine, Baltimore, MD, USA. [4]Sanaria Inc., Rockville, MD, USA. [5]Division of Microbiology and Infectious Diseases, Parasitology and International Programs Branch, NIAID, NIH, Bethesda, MD, USA. [6]These authors contributed equally: Kim C. Williamson, Kirsten E. Lyke. ✉e-mail: kim.williamson@usuhs.edu; klyke@som.umaryland.edu

