## [Peer Review File · Nature Communications]

Repeat controlled human *Plasmodium falciparum* infections delay bloodstream patency and reduce symptomsReviewers' Comments:

Reviewer #1:

Remarks to the Author:

In this manuscript, Ferrer and colleagues describe the parasitological, clinical and immunological course of repeated mosquito bite-induced CHMIs in malaria naïve adults. They observe a small but significant increase in per-patent period, though no sterile protection and no significant change in blood-stage multiplication rate. Frequency, breadth and severity of adverse events decrease across sequential CHMIs, despite participants remaining parasitaemic longer (assuming a fixed liver-stage duration) and attaining more or less similar parasite densities prior to treatment. Immunologically there was a gradual increase in anti-PfCSP titer, CXCR5-expressing CD4+ and CD8+ T cells (including PD1-expressing CD8+ T cells) and peak circulating IL-10 levels and a decrease in CXCR6-expressing CD4+ and CD8+ T cells and peak circulating IFN γ levels.

To my knowledge, this is the first study in the modern CHMI era to assess repeated CHMIs in malaria-naïve adults, despite this topic frequently being discussed within the community – for which the authors are to be highly commended. This study adds a valuable piece to the far from complete jigsaw puzzle that is malaria immunology.

My comments relate to descriptions and interpretation:

Please (at least concisely) describe the conduct of the CHMI's in the M&M, which after all form the basis of the entire study. Some important details (NF54 strain, 5 infected mosquitoes, timeframe) are mentioned (only) in the legend of Fig 1, but various other methodological details that can impact inflammation/symptoms are not described anywhere (except in the accompanying study protocol): Upon what criteria was treatment initiated? (i.e. TBS+ rather than usPCR+; TBS were read in a systematic way/fixed volume of ~1 uL blood assessed -> detection limit ~2 par/uL). What is detection threshold of usPCR? Were only symptoms prior to start of treatment taken into account, or also symptoms post-treatment? (Protocol: "with ~45% of individuals being diagnosed prior to malaria symptoms (personal experience, UMD)", but not clear from Suppl fig 2 if that was also the case for CHMI's in this trial*, as 1st symptom and 1st TBS+ frequently listed on the same day). What antimalarial was used to treat and was this consistently used? (Protocol allows A/L or A/P, but fast-acting A/L results in stronger systemic inflammatory symptoms during treatment than slow-acting A/P, potentially also resulting in stronger rebound tolerance). What symptomatic treatment was offered (acetaminophen, nsaid)? Were participants recommended to take these presumptively, or only once they developed symptoms?

*did the ratio of pre-treatment to post-treatment symptoms change over the course of 4 CHMIs (despite or presumably due to the increase in pre-patent period)?

As a formality, please state the UMB IRB's approval number for the underlying clinical trial. Consider moving Suppl Fig 1 into the main manuscript and please at least briefly refer to the overall design of the trial in the M&M, e.g. that participants could be replaced halfway through (otherwise it is perplexing why some participants did not undergo mock CHMI; also good to mention that no withdrawals were due to intolerable symptoms, which would otherwise bias the overall interpretation) and that (fresh) naïve controls were included for each CHMI. Did the pre-patent period indeed remain stable across each respective set of naïve controls (as verification that the stringency of challenge did not differ from one CHMI to the next) and did the pre-patent period start to diverge (significantly) between the main participants and each set of naïve controls across sequential CHMIs?

Personally, I find the use of the word "patency" on its own (especially in the Abstract) to mean day-of-patency/time-to-patent-parasitaemia etc. unintuitive and confusing; my first assumption was that "patency" referred to a dichotomous outcome (i.e. participants who did or did not develop patent parasitaemia), which actually results in the opposite interpretation to that intended!; this is only

clarified somewhere within the Results section.

Fig 2b could be clarified slightly to indicate which colored lines overlap at each point; also indicating CHMIs using the same colors which are used in the other panels to indicate individual participants is slightly confusing.

Could you mention (absence of) thrombocytopenia in the Results section and possibly discuss? This is quite remarkable, particularly during the earlier more symptomatic CHMIs, given that it usually a very sensitive parameter of both naturally-acquired and controlled infections, associated with inflammation/endothelial cell activation and might have been expected to decrease in severity over the course of sequential CHMIs.

Why were the two replacement participants excluded from the flow cytometry analyses when they are included in the other immunology analyses? And presumably cryopreserved samples from all timepoints per participant were assayed simultaneously to avoid batch effects?

Did you also perform stimulation experiments to assess cellular responses to sporozoites and/or blood-stage parasites??

Why was post-mock challenge taken as baseline for the circulating cytokine measurements? Did you observe significant changes pre/during/post mock challenge? Of the three endogenous pyrogens, IL-6 and TNF α behave more or less as you would expect and correspond with dynamics of fever across the CHMIs, but the most important one, IL-1 β looks bizarre (even assuming the baseline measurements were abnormally high) – what is going on there??

In general, the trend in immunological (especially flow cytometry) dynamics from CHMI 1 through 3 often seems to be reversed during CHMI 4, which does not appear to be caused by missing values. Can you explain this?

Discussion:

I would argue that observed increase in pre-patent period (on average \sim 2 days between CHMI #1 and #4, corresponding with \sim 1 blood-stage multiplication cycle, so in P.f. a \sim 10-20 reduction in liver-to-blood inoculum (even less if some reduction in blood-stage multiplication rate is also factored in), so only 2-3 fold reduction in liver load per sequential CHMI) is relatively (surprisingly?) small, especially for homologous re-infections. By comparison, amongst participants undergoing immunization by patent sporozoite inoculation by mosquito-bite under CQ-prophylaxis, approximately two-thirds remain PCR negative during already their second immunization, indicating that a single exposure (albeit to generally 15 infectious bites) is sufficient to induce sterilizing pre-erythrocytic immunity to homologous re-infection. A similar phenomenon, but slightly lower percentages, is observed with participants immunized with PfSPZ-CVac. In the only trial I am aware of in which participants underwent immunisation by 3x 5 infectious mosquito bites [Bijker et al. J Infect Dis 2014], i.e. most comparable to the exposure conditions in your trial, 3 out of 10 were still already sterilely protected by the second immunisation. Interval between exposures may of course play a role (1 month in the immunization trials, 1 year in yours), but 4 out of 6 participants immunized by Roestenberg et al. with 3x 15 mosquito bites were still sterilely protected \sim 2.5 years later. Judging by the power calculation described in your trial Protocol, you were also expecting to see better protection?

Could the authors comment on (potential causes for) these discrepancies and how they may further our understanding of malaria immunity?

The authors are correct to discuss that their blood-stage infections are much shorter/lower than in either naturally-acquire infections and malaria therapy, which may explain their observed lack of blood-stage immunity. (There have been some nice reviews [e.g. Collins et al. AmJTMH 1999] of the outcomes of repeated malaria therapy infections which, although they do not contain any multiparameter flow cytometry or single cell omics, do shed some interesting light on the development

of pre- erythrocytic, blood-stage and anti-disease immunity and could possibly be discussed in a little more detail.). What might also be interesting discussing in this context, is the potential suppressive effect of parasitaemia on (the development of) pre-erythrocytic immunity. If repeated CHMI (treated at first TBS+) falls somewhere midway on the blood-stage exposure spectrum between naturally-acquired infections/malaria therapy on the one side and immunisation under CQ on the other side, might that be an explanation for the respective development of sterilising pre-erythrocytic immunity observed in these settings?

Reviewer #2:
Remarks to the Author:

This is an interesting paper that shows that a degree of clinical and parasitological immunity against *P. falciparum* malaria in humans can be achieved after as few as two infections with a homologous parasite strain.

While the data are interesting and the study well-executed, this does not present a conceptual advance as similar data have been published back in 1999 which the authors have not referenced.

The reviewer refers the authors to the papers below.

Primary reference - Collins and Jeffery 1999 (PMID: 10432041) and a related relevant manuscript Collins and Jeffery 1999 (PMID: 10432044)

The work is largely descriptive and there is little to fault. I dispute the claim that this is shown for the first time. The authors have added contemporary immunology and no doubt conducted a "cleaner" and better controlled study - but the principle has been observed previously.

The major flaw is the failure to reference similar work as mentioned above. The limitations of the study are acknowledged - the small sample sizes limit the conclusions. The methodology is sound and meets the expected standards in the field. There is enough detail provided in the methods for the work to be reproduced.

Minor comments

In the abstract - I believe the authors meant indicate "time to bloodstream patency" - line 7

Reviewer #3:
Remarks to the Author:

The study by Ferrer et al. investigates the impact of repeated *Plasmodium falciparum* exposure (via mosquito inoculation) on various factors associated with pathogenesis and the development of protective immunity in a controlled human malaria infection (CHMI) model using malaria naïve subjects. The study found that as few as 2 exposures increased the time to bloodstream patency and reduced the severity of symptoms upon subsequent challenge. Importantly, blood stage multiplication rate was unaffected by repeated exposures suggesting liver stage immunity was likely responsible for this clinical observation. In characterizing the immune responses following exposures, the authors found that patency day was correlated with anti-CSP IgG and circulating effector CD8+ T cells. A trend was also observed, that over multiple exposures, there was a reduction in circulating CXCR6+ CD8 and CD4 T cells, suggesting potential retention of those cells to the liver. This would be consistent with

liver TRM as a correlate of protection, as has been described in mouse models. Importantly, the observed lack of sterilizing immunity generated with these multiple exposures using the same parasite strain suggests that the slow acquisition of immunity is likely due to properties of the host-parasite interaction rather than solely strain polymorphisms in the field. The study, which is a somewhat heroic undertaking, is well done and of high interest, as explanations for lack of sterilizing immunity following repeated malaria exposures continue to confound the field. However, several points should be addressed:

1. It is noteworthy that time to PCR positivity was not significantly increased after multiple CHMI, although time to patency increased. I think it worth including some discussion of what the authors think about this outcome in relation to their contention that their data correlate with pre-erythrocytic immunity.
2. PfGLURP should be defined prior to its usage as an acronym (line 102).
3. Line 136 in relation to fig 5A, the authors indicate "the total number of CD4+ T cells expressing CxCR5 is increased..." However, only frequencies are show; not absolute numbers. This issue recurs in the discussion and should be rectified for precision, only frequencies are measured in this work..
4. Figure 1 displays clinical changes over time in individual subjects by the use of different colors, making it easier to identify trends among individual subjects. However, in many of the later figures and supplementals, this pattern is abandoned making it challenging to correlate immunological trends with clinical trends. It would be good to continue to identify specific subjects in subsequent figures when appropriate.
5. It is interesting that many of the figures identify significant immunological changes from CHMI 1 vs. CHMI 3, but not necessarily CHMI 4. Do the authors have thoughts on why this may be? Or is it simply an anomaly based on experimental power?
6. Figure 4B seems to be out of place; being discussed after figures 5 and 6 in order in the results section.
7. Previous studies in mice (PMID: 20619696 and others) have demonstrated that repeated antigen exposures separated by time can modulate T cell functionality. This issue seems germane to the discussion.

Reviewer #4:

Remarks to the Author:

Responses to Reviewer Comments

Reviewer #1 (Remarks to the Author):

In this manuscript, Ferrer and colleagues describe the parasitological, clinical and immunological course of repeated mosquito bite-induced CHMIs in malaria naive adults. They observe a small but significant increase in per-patent period, though no sterile protection and no significant change in blood-stage multiplication rate. Frequency, breadth and severity of adverse events decrease across sequential CHMIs, despite participants remaining parasitaemic longer (assuming a fixed liver-stage duration) and attaining more or less similar parasite densities prior to treatment. Immunologically there was a gradual increase in anti-PfCSP titer, CXCR5-expressing CD4+ and CD8+ T cells (including PD1-expressing CD8+ T cells) and peak circulating IL-10 levels and a decrease in CXCR6-expressing CD4+ and CD8+ T cells and peak circulating IFN γ levels.

To my knowledge, this is the first study in the modern CHMI era to assess repeated CHMIs in malaria-naïve adults, despite this topic frequently being discussed within the community – for which the authors are to be highly commended. This study adds a valuable piece to the far from complete jigsaw puzzle that is malaria immunology.

My comments relate to descriptions and interpretation:

Please (at least concisely) describe the conduct of the CHMI's in the M&M, which after all form the basis of the entire study. Some important details (NF54 strain, 5 infected mosquitoes, timeframe) are mentioned (only) in the legend of Fig 1, but various other methodological details that can impact inflammation/symptoms are not described anywhere (except in the accompanying study protocol):

Response: We have updated the Methods, lines 336-338, 351-366, with a description of CHMI and cited data (Friedman-Klabanoff et al, AJTMH 2019) for full description describing the technique in full.

Upon what criteria was treatment initiated? (i.e. TBS+ rather than usPCR+; TBS were read in a systematic way/fixed volume of ~1 uL blood assessed -> detection limit ~2 par/uL).

Response: Following protocol, 10 μ L of whole blood are spread onto a 1 x 2 cm grid and 5 passes equate to ~1 μ L volume. The limit of detection is 2 unquestionable parasites per 1 μ L volume. Treatment occurred following TBS positivity (lines 360-364). We have cited Friedman-Klabanoff et al, AJTMH 2019 for full description.

What is detection threshold of usPCR?

Response: The limit of detection is approximately 16-20 parasites per milliliter. We have cited Friedman-Klabanoff et al, AJTMH 2019 and Adams et al, Malaria Journal, vol 14, 2015 for full description.

Were only symptoms prior to start of treatment taken into account, or also symptoms post-treatment? (Protocol: "with ~45% of individuals being diagnosed prior to malaria symptoms (personal experience, UMD)", but not clear from Suppl fig 2 if that was also the case for CHMI's in this trial*, as 1st symptom and 1st TBS+ frequently listed on the same day).

Response: During the earlier phase of CHMIs that utilized TBS exclusively, as the diagnostic, chloroquine was utilized and rebound symptomatology was frequently noted. Our center has moved exclusively to usPCR and alternative medications as anti-malarial therapy. However, for the repetitive challenge study, we reverted to TBS for diagnosis, to enable an immune response to develop but maintaining safety parameters in follow-up. Solicited and unsolicited AEs were monitored from Days 6-28 (pre and post-therapy). We have included wording for clarification (Lines 358-359).

What antimalarial was used to treat and was this consistently used? (Protocol allows A/L or A/P, but fast-acting A/L results in stronger systemic inflammatory symptoms during treatment than slow-acting A/P, potentially also resulting in stronger rebound tolerance).

Response: Our primary course of therapy was Malarone which proves easy to administer by directly observed therapy owing to the once a day formulation. As back-up, CoArtem was used in the event of intolerance (nausea/vomiting) to Malarone. We have added wording to explain this approach (Lines 364-366).

What symptomatic treatment was offered (acetaminophen, nsaid)?

Response: Per protocol, ibuprofen or acetaminophen were offered for symptomatic relief.

Were participants recommended to take these presumptively, or only once they developed symptoms?

*did the ratio of pre-treatment to post-treatment symptoms change over the course of 4 CHMIs (despite or presumably due to the increase in pre-patent period)?

Response: Participants were informed of the presence of symptomatic treatment options and were encouraged to request medication if symptoms dictated. However, medication was not provided preemptively. We had next to no post-treatment symptoms reported so we cannot ascertain any change over the course of follow-up. However, as reported in the Results section 'Reduction of reported symptoms', there was a dramatic drop in pre-treatment symptoms reported as related to malaria.

As a formality, please state the UMB IRB's approval number for the underlying clinical trial.

Response: For the reviewer, the University of Maryland IRB approval number was HP-00069877, but for the purposes of publication, we have incorporated the ClinicalTrials.gov#: NCT03014258, line 336.

Consider moving Suppl Fig 1 into the main manuscript and

Response: Suppl Fig 1 is now Fig 2

please at least briefly refer to the overall design of the trial in the M&M, e.g. that participants could be replaced halfway through (otherwise it is perplexing why some participants did not undergo mock CHMI; also good to mention that no withdrawals were due to intolerable symptoms, which would otherwise bias the overall interpretation) and that (fresh) naïve controls were included for each CHMI.

Response: Thank you for this suggestion and we have added additional wording to the Methods section, Lines 98-99 and 355-358.

Did the pre-patent period indeed remain stable across each respective set of naïve controls (as verification that the stringency of challenge did not differ from one CHMI to the next) and did the pre-patent period start to diverge (significantly) between the main participants and each set of naïve controls across sequential CHMIs?

Response: One of the critical and important features of CHMI is embedding infectivity controls. As such, they can also serve as barometers for inter-assay (in this case CHMI) variability. All ICs clustered together with pre-patent periods matching that of the infective CHMI1 (mean – 11.7 days, range 9-14 days vs mean 11.46 days, range 9-12, respectively). Refer to Lines 81-82

Personally, I find the use of the word “patency” on its own (especially in the Abstract) to mean day-of-patency/time-to-patent-parasitaemia etc. unintuitive and confusing; my first assumption was that “patency” referred to a dichotomous outcome (i.e. participants who did or did not develop patent parasitaemia), which actually results in the opposite interpretation to that intended!; this is only clarified somewhere within the Results section.

Response: Patency has been replaced with time to patency or delayed patency throughout the abstract and manuscript.

Fig 2b could be clarified slightly to indicate which colored lines overlap at each point; also indicating CHMIs using the same colors which are used in the other panels to indicate individual participants is slightly confusing.

Response: The lines in Fig 2b been revised by including dashed lines and different colors.

Could you mention (absence of) thrombocytopenia in the Results section and possibly discuss? This is quite remarkable, particularly during the earlier more symptomatic CHMIs, given that it usually a very sensitive parameter of both naturally-acquired and controlled infections, associated with inflammation/endothelial cell activation and might have been expected to decrease in severity over the course of sequential CHMIs.

Response: We did note a dip in individual platelet levels coinciding with the onset of parasitemia. However, nearly all participants' levels remained within the “normal” range. Only a minority of participants had transient dips of platelets to a category of mild thrombocytopenia (below 149,999 cells/mm³). As the normal range extended from 150,000-414,000 cells/mm³, large shifts could occur while remaining within the acceptable range. We have noted this within the Results (Lines 105-106).

Why were the two replacement participants excluded from the flow cytometry analyses when they are included in the other immunology analyses? And presumably cryopreserved samples from all timepoints per participant were assayed simultaneously to avoid batch effects?

Response: Yes, the timepoints for all the subjects tested were assayed simultaneously, unfortunately the 2 subjects that did not complete the mock CHMI were not included in the assay.

Did you also perform stimulation experiments to assess cellular responses to sporozoites and/or blood-stage parasites??

Response: These are the next set of planned experiments. To carefully compare responses to both sporozoite and blood-stage parasites, a number of controls are needed making it quite a large experiment that we feel is beyond the scope of this initial manuscript.

Why was post-mock challenge taken as baseline for the circulating cytokine measurements?

Did you observe significant changes pre/during/post mock challenge?

Response: As expected there was random variation between time points D1, D6, D8, D13, D20 of the mock and D1 of CHMI 1, but there was no discernable pattern. To compensate for this fluctuation, we used the average value through the mock. To better depict the actual cytokine levels, we have included a heatmap of the average cytokine concentrations for each timepoint, including the mock, as Supplementary Figure 6.

Of the three endogenous pyrogens, IL-6 and TNF behave more or less as you would expect and correspond with dynamics of fever across the CHMIs, but the most important one, IL-1b looks bizarre (even assuming the baseline measurements were abnormally high) – what is going on there??

Response: We were also initially surprised by this. It was not due to a failure of detection, since the Luminex standard curve detected levels up to 1400 pg/ml and IL-1b was detected using the same kit to analyze cytokine levels after LPS stimulation of naïve PBMCs. It was also not due to high levels following the mock challenge, although two subjects did have higher levels prior to the mock (D1) these levels decreased by D6. We have added a heat map of the cytokine concentrations as Supplementary Figure 6 for clarification. In checking the literature more carefully we found that during uncomplicated malaria infections in contrast to IL-6 or TNF there were minimal to increases in IL-1b levels (PMIDs: 28122790, 23967342, 21687657). Conversely, in vitro stimulation of PBMCs with a parasite extract does induce IL-1b (PMIDs: 17015763). In the in vitro experiments the standard dose of 10^6 iRBC/ml is higher than that obtained during the CHMI ($10^{4.8}$ - $10^{3.8}$ parasites/ml). Indeed, during severe malaria IL-1b levels do increase consistent with a dose dependent response. IL-1b is produced as an inactive pro-peptide and additional signals, such as those from the inflammasome, are needed to activate cleavage and secretion. Higher parasitemia may be needed to stimulate enough IL-1b release for detection in the plasma, compared to IL-6 or TNF. We added the lack of IL-1b in other field studies of uncomplicated malaria to the discussion, line 287-290.

In general, the trend in immunological (especially flow cytometry) dynamics from CHMI 1 through 3 often seems to be reversed during CHMI 4, which does not appear to be caused by missing values. Can you explain this?

Response: It is a consistent trend and is also observed in the anti-CSP antibody titers. It suggests the presence of negative feedback that limits activation after some level response has been achieved. The increase in the inhibitory ligand PD-1 expression on T cells and/or the previously reported increase in V delta 1 gamma T cells at later CHMIs could modulate the response. Further work is needed to directly evaluate this. We have added this to the discussion line 258-259.

Discussion:

I would argue that observed increase in pre-patent period (on average ~2 days between CHMI #1 and #4, corresponding with ~1 blood-stage multiplication cycle, so in P.f. a ~10-20 reduction in liver-to-blood inoculum (even less if some reduction in blood-stage multiplication rate is also factored in), so only 2-3 fold reduction in liver load per sequential CHMI) is relatively (surprisingly?) small, especially for homologous re-infections. By comparison, amongst participants undergoing immunization by patent sporozoite inoculation by mosquito-bite under CQ-prophylaxis, approximately two-thirds remain PCR negative during already their second immunization, indicating that a single exposure (albeit to generally 15 infectious bites) is sufficient to induce sterilizing pre-erythrocytic immunity to homologous re-infection. A similar phenomenon, but slightly lower

percentages, is observed with participants immunized with PfSPZ-CVac. In the only trial I am aware of in which participants underwent immunisation by 3x 5 infectious mosquito bites [Bijker et al. J Infect Dis 2014], i.e. most comparable to the exposure conditions in your trial, 3 out of 10 were still already sterilely protected by the second immunisation. Interval between exposures may of course play a role (1 month in the immunization trials, 1 year in yours), but 4 out of 6 participants immunized by Roestenberg et al. with 3x 15 mosquito bites were still sterilely protected ~2.5 years later. Judging by the power calculation described in your trial Protocol, you were also expecting to see better protection?

Could the authors comment on (potential causes for) these discrepancies and how they may further our understanding of malaria immunity? What might also be interesting discussing in this context, is the potential suppressive effect of parasitaemia on (the development of) pre-erythrocytic immunity. If repeated CHMI (treated at first TBS+) falls somewhere midway on the blood-stage exposure spectrum between naturally-acquired infections/malaria therapy on the one side and immunisation under CQ on the other side, might that be an explanation for the respective development of sterilising pre-erythrocytic immunity observed in these settings?

Response: We were also anticipating a more robust protective response to homologous challenges. The actual findings highlight how well adapted the parasite is to the human immune response and the importance of carefully studying this response in a controlled setting in humans.

The interval between exposures undoubtedly plays a role. Referring to the PfSPZ CVac model, high level protection was established by reducing the interval of exposure to PfSPZ to Days 1, 8, and 29 (Mordmueller et al, Nature, 2017), implying that short bursts of exposure at reduced intervals may “stack” immunity. This is further evidenced by our finding that an analogous delivery schema to Ty21A of Days 1, 3, 5, 7 “stacked” PfSPZ Vaccine delivery provided better protection than three doses administered at 8-week intervals (Lyke et al, CID, 2021). The 6-9-month CHMI intervals in our repeat CHMI study were largely dictated by the annual funding limitations, so we did not have much flexibility. Within these constraints we strived to deliver a predictable CHMI at 6 mo intervals. However, as you indicate it is also possible that the presence of blood stage parasites could influence the response, as suggested in mouse malaria models (Keitany et al, Cell Reports 2016) We have added this to the discussion, lines 212-229. Our future work includes analysis of genomic responses via CITE-seq or other platforms to further define the factors contributing to the partial response.

The authors are correct to discuss that their blood-stage infections are much shorter/lower than in either naturally-acquire infections and malaria therapy, which may explain their observed lack of blood-stage immunity. (There have been some nice reviews [e.g. Collins et al. AmJTMH 1999] of the outcomes of repeated malaria therapy infections which, although they do not contain any multiparameter flow cytometry or single cell omics, do shed some interesting light on the development of pre- erythrocytic, blood-stage and anti-disease immunity and could possibly be discussed in a little more detail.).

Response: We have added more information about the malaria fever therapy trials for syphilis patients to the introduction (Lines 36-41) and included the 1999 reference in addition to the 2001 reference to this work we originally included. As mentioned the extremely high parasitemia attained and maintained for weeks in that work differentiates it from our clinical trial, but they do demonstrate partial immunity against homologous blood stage parasites, including a reduction in peak parasitemia and fever days, is possible after prolonged exposure to asexual parasites. The subjects that were rechallenged with sporozoites also remained susceptible.

Reviewer #2 (Remarks to the Author):

This is an interesting paper that shows that a degree of clinical and parasitological immunity against P. falciparum malaria in humans can be achieved after as few as two infections with a homologous parasite strain.

While the data are interesting and the study well-executed, this does not present a conceptual advance as similar data have been published back in 1999 which the authors have not referenced.

The reviewer refers the authors to the papers below.

Primary reference - Collins and Jeffery 1999 (PMID: 10432041) and a related relevant manuscript Collins and Jeffery 1999 (PMID: 10432044)

The work is largely descriptive and there is little to fault. I dispute the claim that this is shown for the first time.

The authors have added contemporary immunology and no doubt conducted a "cleaner" and better controlled study - but the principle has been observed previously.

The major flaw is the failure to reference similar work as mentioned above. The limitations of the study are acknowledged - the small sample sizes limit the conclusions.

The methodology is sound and meets the expected standards in the field. There is enough detail provided in the methods for the work to be reproduced.

Response: As indicated in the response to reviewer 1 we have added more information about the malaria fever therapy trials carried out from the 1940s to 60s to the introduction, lines 36-41 and included the 1999 references in addition to the 2001 reference to this work that we originally included. This prior work focused on the parasitemia and clinical findings with little analysis of the immune response. However, to not overstate our study we removed first time throughout the text.

Minor comments

In the abstract - I believe the authors meant indicate "time to bloodstream patency" - line 7

Response: Thanks for catching this. It has been corrected, lines 7-9.

Reviewer #3 (Remarks to the Author):

The study by Ferrer et al. investigates the impact of repeated Plasmodium falciparum exposure (via mosquito inoculation) on various factors associated with pathogenesis and the development of protective immunity in a controlled human malaria infection (CHMI) model using malaria naïve subjects. The study found that as few as 2 exposures increased the time to bloodstream patency and reduced the severity of symptoms upon subsequent challenge. Importantly, blood stage multiplication rate was unaffected by repeated exposures suggesting liver stage immunity was likely responsible for this clinical observation. In characterizing the immune responses following exposures, the authors found that patency day was correlated with anti-CSP IgG and circulating effector CD8+ T cells. A trend was also observed, that over multiple exposures, there was a reduction in circulating CXCR6+ CD8 and CD4 T cells, suggesting potential retention of those cells to the liver. This would be consistent with liver TRM as a correlate of protection, as has been described in mouse models. Importantly, the observed lack of sterilizing immunity generated with these multiple exposures using the same parasite strain suggests that the slow acquisition of immunity is likely due to properties of the host-parasite interaction rather than solely strain polymorphisms in the field. The study, which is a somewhat heroic undertaking, is well done and of high interest, as explanations for lack of sterilizing immunity following repeated malaria exposures continue to confound the field. However, several points should be addressed:

1. It is noteworthy that time to PCR positivity was not significantly increased after multiple CHMI, although time to patency increased. I think it worth including some discussion of what the authors think about this outcome in relation to their contention that their data correlate with pre-erythrocytic immunity.

Response: Yes, although there was a slight increase in the time to PCR positivity it was not significant. This is likely due to the sensitivity of the PCR and that we only collected daily samples which limited finer resolution of the initial release and growth rate needed to assess more subtle delays in patency. This has been added to the results section, lines 86-88.

2. PfGLURP should be defined prior to its usage as an acronym (line 102).

Response: Done, line 115

3. Line 136 in relation to fig 5A, the authors indicate "the total number of CD4+ T cells expressing CxCR5 is increased..." However, only frequencies are shown; not absolute numbers. This issue recurs in the discussion and should be rectified for precision, only frequencies are measured in this work.

Response: This has been corrected in line 149 and the rest of the text checked.

4. Figure 1 displays clinical changes over time in individual subjects by the use of different colors, making it easier to identify trends among individual subjects. However, in many of the later figures and supplementals, this pattern is abandoned making it challenging to correlate immunological trends with clinical trends. It would be good to continue to identify specific subjects in subsequent figures when appropriate.

Response: The colors for the subjects have been added to the immunophenotyping figures.

5. It is interesting that many of the figures identify significant immunological changes from CHMI 1 vs. CHMI 3, but not necessarily CHMI 4. Do the authors have thoughts on why this may be? Or is it simply an anomaly based on experimental power?

Response: A plateau or decrease at CHMI 4 is a consistent trend and is also observed in the anti-CSP antibody titers suggesting the presence of negative feedback that limits activation once some level has been achieved. The increase in the inhibitory ligand PD-1, the previously reported increase in V delta 1 gamma T cells at later CHMIs and/or changes associated with the decrease in IFN γ could modulate the response. We also note that CHMI were performed at months 2, 9, 14 and 23 with the longest interval occurring between iCHIM3 and 4 (9 months). Further work is needed to directly evaluate this. We have added this to the discussion line 223, 258-259.

6. Figure 4B seems to be out of place; being discussed after figures 5 and 6 in order in the results section.

Response: Our idea was to show the responses that correlated with the time to patency together in one figure. We can separate this, but would prefer not to.

7. Previous studies in mice (PMID: 20619696 and others) have demonstrated that repeated antigen exposures separated by time can modulate T cell functionality. This issue seems germane to the discussion.

Response: This and additional references have been added to the discussion of the changes in the immunophenotypes of the T cells following repetitive stimuli, line 256-257.

Reviewers' Comments:

Reviewer #1:

Remarks to the Author:

I thank the authors for taking the time to respond to all queries and to improve the interpretation of their manuscript. I have no further queries or concerns.

I suspect that some of the remaining differences between their CHMI model and that at other centers (e.g. relative absence of thrombocytopenia and post-treatment symptoms) may be related in part to differences in pre-patent period between these models, which in turn reflects liver-to-blood inoculum. The relative size of this sudden introduction of pathogenic micro-organisms into circulation may well determine the subsequent pattern of systemic inflammatory responses.

Reviewer #3:

Remarks to the Author:

Our comments were addressed in a satisfactory manner.